# Adaptive Computation Pruning for the Forgetting Transformer

**Zhixuan Lin**[*]
Mila - Quebec AI Institute
Université de Montréal
zxlin.cs@gmail.com

**Johan Obando-Ceron**
Mila - Quebec AI Institute
Université de Montréal
jobando0730@gmail.com

**Xu Owen He**
MakerMaker AI
owen.hexu@gmail.com

**Aaron Courville**
Mila - Quebec AI Institute
Université de Montréal
courvila@mila.quebec

## Abstract

The recently proposed Forgetting Transformer (FoX) incorporates a forget gate into softmax attention and has shown consistently better or on-par performance compared to the standard RoPE-based Transformer. Notably, many attention heads in FoX tend to forget quickly, causing their output at each timestep to rely primarily on local context. Based on this observation, we propose Adaptive Computation Pruning (ACP) for FoX, a method that dynamically prunes computations involving input-output dependencies that are strongly decayed by the forget gate. In particular, our method performs *provably safe* pruning via a dynamically set pruning threshold that guarantees the pruned attention weights are negligible. We apply ACP to language model pretraining with FoX and show it consistently reduces the number of FLOPs and memory accesses in softmax attention by around 70% across different model sizes and context lengths, resulting in a roughly 50% to 70% reduction in attention runtime (or a 2–3$\times$ speedup) and a roughly 10% to 40% increase in end-to-end training throughput. Furthermore, longer context lengths yield greater computational savings. All these speed improvements are achieved *without any performance degradation*. Our code is available at https://github.com/zhixuan-lin/forgetting-transformer.

## 1 Introduction

Transformers (Vaswani et al., 2017) have quadratic time complexity with respect to context length, resulting in significant computational costs over long sequences. The recently proposed Forgetting Transformer (FoX) (Lin et al., 2025) features a modified softmax attention mechanism with a forget gate, which allows some attention heads to downweight distant dependencies and focus mainly on the local context. FoX has been shown to consistently achieve better or on-par performance compared to the standard RoPE-based (Su et al., 2024) Transformer in various tasks, including long-context language modeling and downstream tasks such as the needle-in-a-haystack test (Kamradt, 2023). It is also compatible with the FlashAttention (Dao, 2024) algorithm, which allows efficient processing of long sequences.

Lin et al. (2025) show that many attention heads in FoX tend to forget quickly. For these heads, the dependencies between distant input-output pairs are extremely weak and can potentially be ignored. Based on this observation, we propose *Adaptive Computation Pruning (ACP)* for FoX, a method that dynamically prunes computations involving input-output dependencies that are strongly decayed by the forget gate. In particular, our method performs *provably safe* pruning via a dynamically set threshold that guarantees the total pruned attention weights are bounded by a hyperparameter $\varepsilon$. In practice, we set $\varepsilon = e^{-10} \approx$

---

[*]Correspondence to Zhixuan Lin.

**Without ACP**

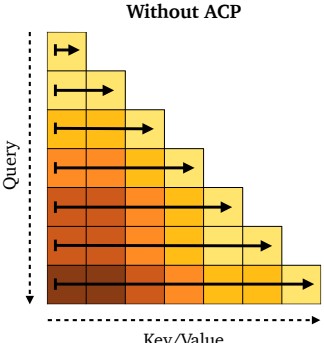

Query

Key/Value

**With ACP**

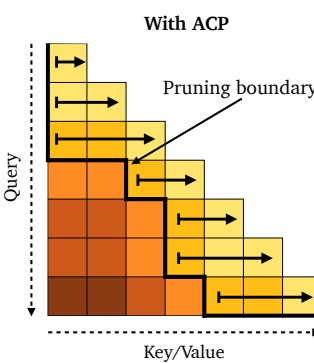

Pruning boundary

Query

Key/Value

Figure 1: **Illustration of Forgetting Attention with and without ACP**. Each cell represents a block in the FlashAttention algorithm. Darker colors indicate more-negative decay bias values and thus stronger decay. The solid arrows indicate the set of blocks that would be visited (in the indicated order) in the FlashAttention iterations.

0.000045, which is effectively negligible. Furthermore, as shown in Figure 1, the decay structure in FoX induces a sliding-window-like pruning pattern, enabling an efficient two-stage implementation. First, we identify a *pruning boundary* across the grid of computations in FlashAttention via a linear-time algorithm. Once the pruning boundary is identified, we restrict the FlashAttention iterations to the remaining blocks, avoiding any wasted computation on pruned dependencies.

We apply ACP to language model *pretraining* with FoX with sizes from 125M to 760M parameters and training context lengths from 4k to 16k tokens. We find that ACP consistently prunes around 70% of the FLOPs and memory accesses in softmax attention across the tested model sizes and context lengths, resulting in a roughly 50% to 70% reduction in attention runtime (or a 2–3× speedup) and a roughly 10% to 40% increase in training throughput. In particular, longer context lengths lead to greater computational savings and speedups. These speed improvements are achieved *without affecting language modeling loss and downstream task performance*. To provide further insight into our method, we conduct a series of analyses such as examining the pruning boundaries and analyzing the distribution of computational savings across different attention heads. Notably, our analysis reveals the existence of "local heads" and "global heads" that are responsible for modeling dependencies of different lengths. Finally, in addition to our current results that focus on applying ACP during *pretraining*, we also discuss how ACP could be used to reduce computation and memory usage for prefilling and decoding during *inference*, along with preliminary results.

## 2 Preliminaries: Forgetting Transformer

This section gives a brief introduction to the Forgetting Transformer and in particular its FlashAttention-based implementation. Throughout this work, we follow Yang et al. (2024) and use notation such as $A_{[m]}$ and $A_{[m][n]}$ to index a block of a matrix (or a vector). For example, for a matrix $A \in \mathbb{R}^{L \times L}$ and block sizes $B_q$ and $B_k$ for the two dimensions of $A$, $A_{[m][n]} \in \mathbb{R}^{B_q \times B_k}$ would be a block of $A$ such that $(A_{[m][n]})_{xy} = A_{ij}$, where $i = (m-1) \cdot B_q + x$ and $j = (n-1) \cdot B_k + y$.

The Forgetting Transformer features a modified softmax attention mechanism with a forget gate, called *Forgetting Attention*. Forgetting Attention takes a sequence of input vectors $(x_i)_{i=1}^L$ and produces a sequence of output vectors $(o_i)_{i=1}^L$. In addition to the usual query/key/value projections $q_i, k_i, v_i = W_q x_i, W_k x_i, W_v x_i \in \mathbb{R}^d$, at each timestep we also compute a scalar forget gate $f_t = \sigma(w_f^\top x_t + b_f) \in \mathbb{R}$, where $\sigma$ is the sigmoid function. The

output of the attention is then

$$o_i = \frac{\sum_{j=1}^{i} F_{ij} \exp(q_i^\top k_j / \sqrt{d}) v_j}{\sum_{j=1}^{i} F_{ij} \exp(q_i^\top k_j / \sqrt{d})} = \frac{\sum_{j=1}^{i} \exp(q_i^\top k_j / \sqrt{d} + D_{ij}) v_j}{\sum_{j=1}^{i} \exp(q_i^\top k_j / \sqrt{d} + D_{ij})}, \tag{1}$$

where $F_{ij} = \prod_{l=j+1}^{i} f_l$ and $D_{ij} = \log F_{ij} = \sum_{l=j+1}^{i} \log f_l$, with $F_{ii} = 1$ and $D_{ii} = 0$ for any $i$. This can be written in matrix form:

$$O = \mathrm{softmax}(QK^\top / \sqrt{d} + D)V \in \mathbb{R}^{L \times d}, \tag{2}$$

where $D \in \mathbb{R}^{L \times L}$ is the *decay bias matrix* containing the $D_{ij}$ factors as its lower triangular entries and $-\infty$ above its main diagonal. $Q, K, V, O \in \mathbb{R}^{L \times d}$ are matrices containing $q_i, k_i, v_i, o_i, i \in \{1, \dots, L\}$ as the rows. For multi-head attention with $H$ heads, we maintain $H$ instances of forget gate parameters $\{w_f^{(h)}\}_{h=1}^{H}$ and $\{b_f^{(h)}\}_{h=1}^{H}$ and compute the forget gate values $\{f_t^{(h)}\}_{h=1}^{H}$ separately for each head. We will omit the $(h)$ superscript throughout this work and assume $d$ represents the dimension of each head.

**$D$ is coordinate-wise monotone**  The matrix $D$ has the following property: for any indices $i, j, x, y$ such that $i \geq x$ and $j \leq y$, we have $D_{ij} \leq D_{xy}$. This is visualized in Figure 1, where darker colors indicate more-negative $D_{ij}$ values. This property, which we call *coordinate-wise monotonicity*, is crucial for developing an efficient pruning algorithm.

**FlashAttention implementation of Forgetting Attention**  The $D$ matrix can be computed as $D = c\mathbf{1}^\top - \mathbf{1}c^\top$, where $c \in \mathbb{R}^L$ contains the cumulative sums $c_i = \sum_{l=1}^{i} \log f_l, i \in \{1, \dots, L\}$ and $\mathbf{1} \in \mathbb{R}^L$ is a vector of all ones. This makes it possible to implement Forgetting Attention with a simple modification to the FlashAttention algorithm.

We briefly describe the forward pass. In FlashAttention, queries are divided into $M$ blocks $\{Q_{[m]} \in \mathbb{R}^{B_q \times d}\}_{m=1}^{M}$ with block size $B_q = L/M$. The keys and values are similarly divided into $N$ blocks $\{K_{[n]}, V_{[n]} \in \mathbb{R}^{B_k \times d}\}_{n=1}^{N}$ with block size $B_k = L/N$. All the computations are then conceptually organized into a $M \times N$ grid, as shown in Figure 1. In standard softmax attention without forget gates, FlashAttention computes the attention logit blocks $S_{[m][n]} = Q_{[m]} K_{[n]}^\top / \sqrt{d}$ in the shared memory (SMEM) of the GPU sequentially across the key/value block dimension $N$ and in parallel across the query dimension $M$ (see Figure 1 left). To implement Forgetting Attention, we only need to additionally load $c_{[m]}$ and $c_{[n]}$ into SMEM, construct $D_{[m][n]} = c_{[m]} \mathbf{1}^\top - \mathbf{1}c_{[n]}^\top$, and compute the modified attention logits $S_{[m][n]} = Q_{[m]} K_{[n]}^\top / \sqrt{d} + D_{[m][n]}$. The rest of the forward pass remains the same as in standard FlashAttention. The backward pass is implemented similarly.

## 3  Adaptive Computation Pruning

We now introduce our method, Adaptive Computation Pruning (ACP). Conceptually, ACP aims to prune all the computations in the term $\exp(q_i^\top k_j / \sqrt{d} + D_{ij}) v_j$ if $D_{ij} < \delta$, where $\delta < 0$ is a dynamically set threshold (explained later). The attention outputs after pruning are given by:

$$o_i = \frac{\sum_{j=1}^{i} \mathbb{1}\{D_{ij} \geq \delta\} \exp(q_i^\top k_j / \sqrt{d} + D_{ij}) v_j}{\sum_{j=1}^{i} \mathbb{1}\{D_{ij} \geq \delta\} \exp(q_i^\top k_j / \sqrt{d} + D_{ij})}, \tag{3}$$

where $\mathbb{1}\{\cdot\}$ is the indicator function that takes 1 if the inner proposition is true and 0 otherwise. The intuition of ACP is as follows. Let $s_{ij} = q_i^\top k_j / \sqrt{d}$ and $U$ be an upper bound of $\{|s_{ij}|\}_{i,j \in \{1,\dots,L\}}$, i.e. $U \geq \max_{i,j \in \{1,\dots,L\}} |s_{ij}|$. Since by definition $D_{ii} = 0$ for any

$i$, if for some $j$, $D_{ij}$ is much smaller than $-2U$, then the corresponding attention weight $A_{ij} = \frac{\exp(s_{ij}+D_{ij})}{\sum_{k=1}^{i} \exp(s_{ik}+D_{ik})} \leq \frac{\exp(s_{ij}+D_{ij})}{\exp(s_{ii}+D_{ii})} = \exp(s_{ij} - s_{ii} + D_{ij}) \leq \exp(2U - D_{ij})$ would be very small, making the contribution of $v_j$ to $o_i$ negligible. And thus the related computations can be safely skipped.

**Safe pruning via a dynamically set threshold**  In practice, we set the threshold $\delta$ dynamically based on an upper bound $U$ of $\{|s_{ij}|\}_{i,j\in\{1,...,L\}}$ and the sequence length $L$ so that the total pruned attention weights $\sum_{j=1}^{L} \mathbb{1}\{D_{ij} < \delta\} A_{ij}$ for any $i$ would be bounded by a small number $\varepsilon > 0$. Concretely, we set $\delta = -2U - \log L + \log \varepsilon$, which achieves the above guarantee (see Appendix A for a proof). We set $\varepsilon = e^{-10} \approx 0.000045$ throughout this work to ensure that the impact of ACP on attention outputs is negligible.[1]

Setting $\delta$ dynamically requires us to know an upper bound $U$ of $\{|s_{ij}|\}_{i,j\in\{1,...,L\}}$. Since $|s_{ij}| \leq \frac{\|q_i\|_2 \|k_j\|_2}{\sqrt{d}}$, we can set $U = \frac{\rho_q \rho_k}{\sqrt{d}}$, where $\rho_q$ and $\rho_k$ are upper bounds of the L2-norms of the queries and keys respectively. We can either obtain $\rho_q$ and $\rho_k$ by explicitly computing the L2-norms of the queries and keys, or directly derive them from the corresponding normalization parameters if QK-norm (Dehghani et al., 2023) is used (see Appendix E).

**Block-level pruning**  In FlashAttention, computations are performed in blocks. Conceptually, these blocks of computation are organized into an $M \times N$ grid as shown in Figure 1, where $M$ is the number of query blocks and $N$ is the number of key and value blocks. Therefore, in practice, ACP operates at the block level and we prune a computation block $(m, n)$ if and only if all entries in $D_{[m][n]}$ are below $\delta$, or equivalently, if the maximum entry of $D_{[m][n]}$ is below $\delta$. Since $D$ is coordinate-wise monotone, the maximum entry of $D_{[m][n]} \in \mathbb{R}^{B_q \times B_k}$ (denoted as $\max(D_{[m][n]})$ in the following) is simply its top-right entry $(D_{[m][n]})_{1,B_k} = (c_{[m]})_1 - (c_{[n]})_{B_k}$. Therefore, we only need to check this entry to determine whether a block should be pruned.[2]

**Two-stage implementation**  Since $D$ is coordinate-wise monotone, it is easy to show that if $\max(D_{[m][n]}) < \delta$ then $\max(D_{[x][y]}) < \delta$ for any $x \geq m$ and $y \leq n$. This means that the set of computation blocks to be pruned constitutes a consecutive region on the lower-left part of the $M \times N$ grid, as shown in Figure 1 (right). In addition, this region is separated from the rest of the grid by a *pruning boundary* that connects the top-left corner and the bottom-right corner of the grid, yielding a sliding-window-like pruning pattern. Based on this observation, we can perform ACP in two stages. First, we identify the pruning boundary. Specifically, for each row $m$, we determine the first computation block $(m, n_m)$ on the right of the pruning boundary on row $m$. In Figure 1 (right), these correspond to the blocks at the start of each arrow. After this is done, for each row $m$, we start the FlashAttention iterations from block $(m, n_m)$ (instead of block $(m, 1)$), and therefore no computations would be wasted on the pruned blocks. Note that if a block is pruned, the kernel also skips the corresponding memory accesses. For example, in the forward pass, if block $(m, n)$ is pruned, the thread block corresponding to query block $Q_{[m]}$ will not load $K_{[n]}$ and $V_{[n]}$. Therefore *ACP reduces both the number of FLOPs and memory accesses.*

---

[1] For reference, the typical relative rounding error for the popular `bfloat16` precision is on the order of 0.001.

[2] If $D_{[m][n]}$ lies on the diagonal of the grid, it is not pruned by default as it contains an entry $D_{ii}$ for some $i$, which by definition is zero.

---

**Algorithm 1** Index search for boundary blocks

---

**Require:** Cumsum of log forget gates $c \in \mathbb{R}^L$, threshold $\delta$, number of query blocks $M$
**Ensure:** $n_m$ be the column index of the boundary block on row $m$ for each $m \in \{1, \dots, M\}$
1: $l \leftarrow 1$
2: **for** $m$ from 1 to $M$ **do**
3:     $D_{\max} \leftarrow -\infty$
4:     **while** $D_{\max} < \delta$ **do**
5:        $D_{\max} = (c_{[m]})_1 - (c_{[l]})_{B_k}$ (This is the top-right and the maximum entry of $D_{[m][l]}$)
6:        $l \leftarrow l + 1$ (We loop until $(m, l)$ is a boundary block)
7:     **end while**
8:     Set $n_m = l$
9: **end for**

---

**Identifying boundary block indices** The final missing piece of ACP is an algorithm to identify the column index $n_m$ of the boundary block on each row $m$. Since $D$ is coordinate-wise monotone, for any two such boundary blocks $(m, n_m)$ and $(x, y_x)$ we have $m \geq x \iff n_m \geq y_x$. This makes it possible to use an efficient linear-time algorithm to identify the boundary block indices, shown in Algorithm 1. This algorithm has a linear complexity of $O(\max(\frac{L}{B_q}, \frac{L}{B_k}))$, compared to the $O(L^2 d)$ quadratic complexity of standard full attention. In practice, we find that boundary index search accounts for only a minimal portion of the total attention kernel runtime (around 2% to 6%; see Appendix D).

## 4 Experiments

In principle, ACP applies to both pretraining and inference (prefilling and decoding). Because pretraining typically saturates available GPUs, reductions in FLOPs or memory accesses achieved by ACP translate directly into shorter wall-clock time. During inference, ACP should deliver similar reductions in FLOPs and memory accesses; however, realizing comparable end-to-end speedups may require additional optimizations to remove other bottlenecks such as kernel-launch overheads. Therefore, this work focuses on the computational savings and wall-clock improvements of ACP for *pretraining*. We defer a discussion of inference-time ACP and some preliminary results to Appendix F.

### 4.1 Experimental setup

Throughout this work, we use the FoX (Pro) architecture introduced in Lin et al. (2025). Following Lin et al. (2025), we do not use RoPE (Su et al., 2024). The Pro architecture enhances the basic LLaMA (Touvron et al., 2023) architecture by incorporating some common components in recurrent sequence models such as QK-norm (Dehghani et al., 2023), output gate, output normalization, and data-dependent token-shift (Peng et al., 2024). For completeness, we also provide results for the FoX (LLaMA) architecture in Appendix C.2, which are similar to the results for FoX (Pro) that we present below.

We train FoX (Pro) models with and without ACP on LongCrawl64 (Buckman, 2024) using the standard language modeling objective. We adopt the three training configurations used in the analysis experiments in Lin et al. (2025), specified as combinations of the number of model parameters and the number of training tokens: 760M-parameter/16B-token, 360M-parameter/7.5B-token, and 125M-parameter/2.7B-token. For each scale, we train the models with three training context lengths: 4k, 8k, and 16k tokens. The rest of the hyperparameters are the same as those in Lin et al. (2025) and are described in detail in Appendix B.

We use the official Forgetting Transformer repository[3] for the implementation. We implement ACP, including the boundary index search algorithm, on top of the official Forgetting Attention kernel in Triton (OpenAI, 2021).

---

[3]https://github.com/zhixuan-lin/forgetting-transformer

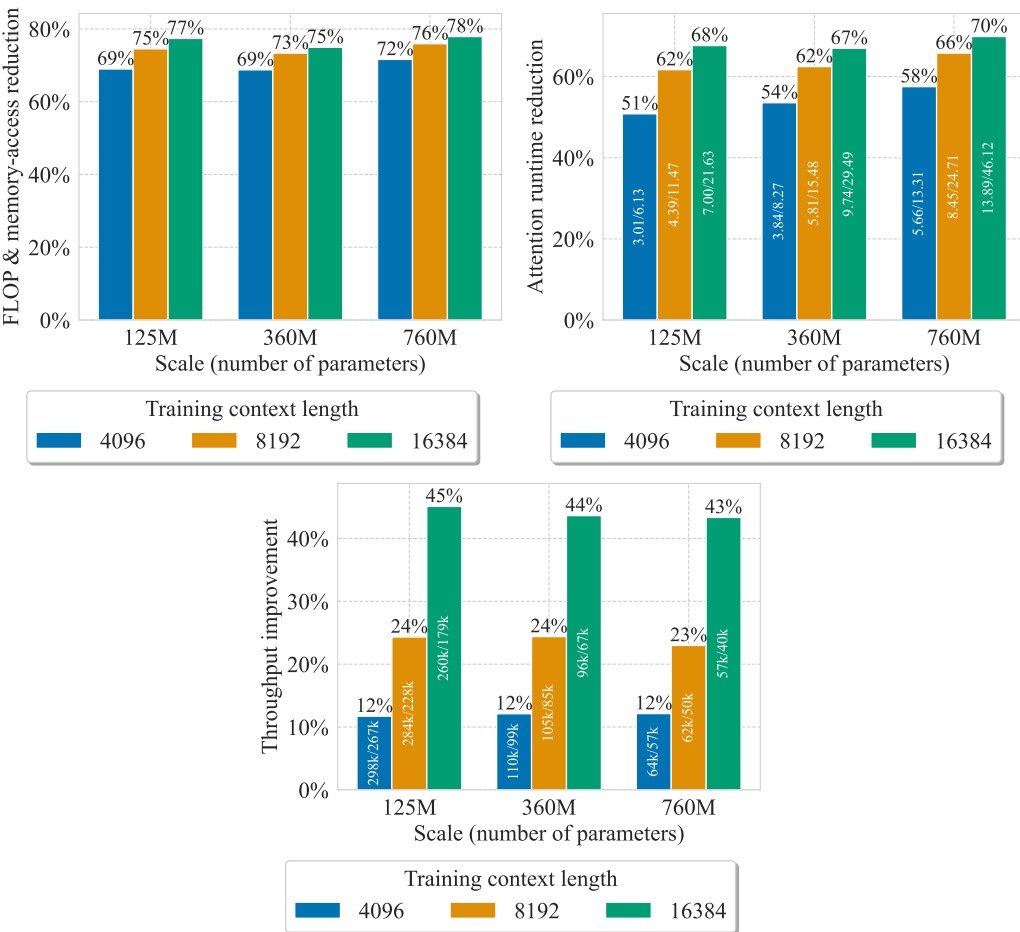

Figure 2: (**left**) Percentage reduction in FLOPs and memory accesses in the attention operation due to ACP. (**right**) Percentage reduction in attention kernel runtime due to ACP. Within each bar we also show the actual runtime with and without ACP in milliseconds. The runtime covers one forward and backward pass on a batch of 0.5M tokens. (**bottom**) Percentage training throughput improvement due to ACP. Within each bar we also show the actual values of training throughput with and without ACP. Throughput is measured in tokens per second. Both the attention kernel runtime and throughput are measured on 4 NVIDIA L40S GPUs.

In the following, training throughputs are measured using the final checkpoints on a subset of the heldout set of LongCrawl64 on 4 NVIDIA L40S GPUs. We find that training throughput typically decreases for a short period at the beginning of training and then plateaus, so our reported numbers using the final checkpoints reflect the throughput during the plateau period. When ignoring sub-leading terms, the percentage reduction in FLOPs and memory accesses in the attention operation can be approximated by the ratio of the number of pruned blocks to the total number of blocks in the FlashAttention grid. We compute this ratio on a subset of the heldout set of LongCrawl64. More details can be found in Appendix B.

## 4.2 Computational savings and speedups

In Figure 2 we show the percentage reduction in FLOPs and memory accesses *in the attention operation*, the percentage reduction in attention kernel runtime, and the percentage improvement in training throughput due to ACP, across different model sizes and training context lengths. As shown in Figure 2, ACP consistently prunes around 70% of the FLOPs

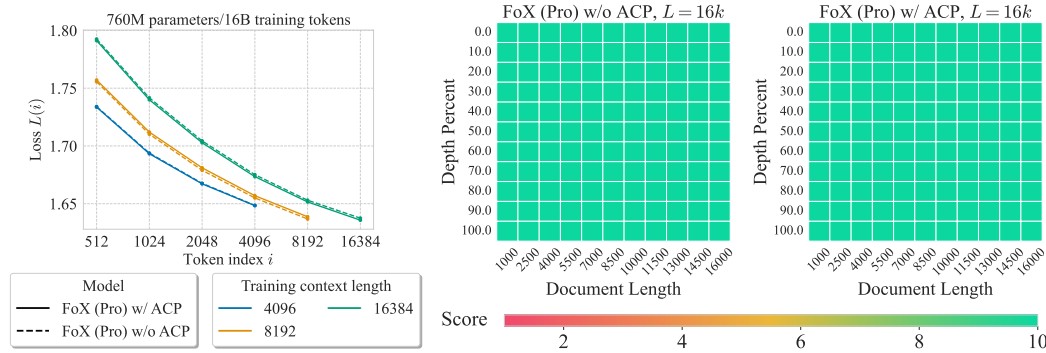

Figure 3: (**left**) Per-token loss given different training context lengths for the 760M-parameter/16B-token setting. This is measured on a 2B-token validation set of the LongCrawl64. At each token index $i$, we report the averaged loss over a window of 101 centered at $i$. (**right**) Easy-mode needle-in-a-haystack results for the 760M-parameter models with a training context length of $L = 16k$ tokens.

| Model | Wiki. ppl↓ | LMB. ppl↓ | LMB. acc↑ | PIQA acc↑ | Hella. acc-n↑ | Wino. acc↑ | ARC-e acc↑ | ARC-c acc-n↑ | COPA acc↑ | OBQA acc-n↑ | SciQA acc↑ | BoolQ acc↑ | Avg ↑ |
|---|---|---|---|---|---|---|---|---|---|---|---|---|---|
| FoX (Pro) w/ ACP, $L = 4k$ | 29.66 | 22.12 | 37.57 | 63.11 | 33.59 | 52.41 | 48.91 | 24.66 | 68.00 | 29.20 | 79.90 | 57.16 | 49.45 |
| FoX (Pro) w/o ACP, $L = 4k$ | 29.98 | 22.32 | 37.65 | 62.84 | 33.38 | 52.72 | 47.94 | 25.60 | 67.00 | 29.60 | 79.70 | 54.22 | 49.06 |
| FoX (Pro) w/ ACP, $L = 8k$ | 28.04 | 23.20 | 38.13 | 60.94 | 33.46 | 51.70 | 48.82 | 24.66 | 67.00 | 28.60 | 80.00 | 60.12 | 49.34 |
| FoX (Pro) w/o ACP, $L = 8k$ | 28.07 | 22.53 | 38.31 | 61.81 | 33.83 | 50.67 | 49.28 | 24.83 | 69.00 | 27.40 | 80.80 | 61.59 | 49.75 |
| FoX (Pro) w/ ACP, $L = 16k$ | 27.96 | 25.16 | 35.77 | 62.35 | 33.79 | 50.83 | 48.02 | 24.23 | 69.00 | 28.20 | 79.50 | 58.93 | 49.06 |
| FoX (Pro) w/o ACP, $L = 16k$ | 28.04 | 24.29 | 36.66 | 62.35 | 33.32 | 48.86 | 48.11 | 25.51 | 71.00 | 27.20 | 82.20 | 56.76 | 49.20 |

Table 1: **Evaluation results on LM-eval-harness.** All models have roughly 760M non-embedding parameters and are trained on roughly 16B tokens on LongCrawl64. "acc-n" means length-normalized accuracy. $L$ is the training context length.

and memory accesses in softmax attention in all cases, resulting in a roughly 50% to 70% reduction in attention runtime (or a 2–3× speedup). These translate into a roughly 10% to 40% increase in end-to-end training throughput. Note that ACP only affects the speed of the attention kernel, whereas training throughput also depends on the latency of other components such as MLPs and RMSNorms. In particular, longer training context lengths lead to greater throughput improvements, because the proportion of FLOPs and memory accesses in softmax attention increases relative to the rest of the network as context length grows. For example, for a 760M-parameter model with a context length of 4k tokens, the attention operation accounts for roughly 16% of the total FLOPs of the model, while for a context length of 16k tokens, it accounts for around 45%.

**ACP does not damage performance** In Figure 3 (left) we show the language modeling loss at different token positions for the 760M-parameter FoX (Pro) models with different training context lengths, with and without ACP. Figure 3 (right) shows the needle-in-a-haystack retrieval results of the 16k-context-length model in Figure 3 (left), following the "easy mode" setup used in Lin et al. (2025) that is suitable for small models without instruction-tuning. Table 1 shows the evaluation results on various downstream tasks from Language Model Evaluation Harness (Gao et al., 2024a) for the models in Figure 3 (left). Additional results can be found in Appendix C.

As shown in these results, the per-token language modeling loss curves with and without ACP almost match exactly (the slight difference is within the expected variance across runs). ACP also does not damage long-context retrieval performance, and the downstream task performances of models with and without ACP are similar. Note that it is well known that evaluation results on downstream tasks can exhibit high variance across training runs (Madaan et al., 2024), so it is impossible to obtain exactly the same results even when training the same model with different seeds.

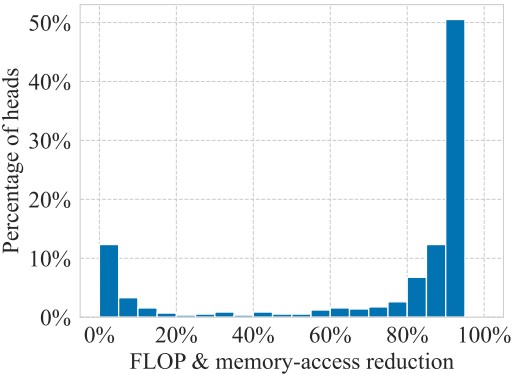

Figure 4: **Distribution of per-head computational savings in a 760M-parameter FoX (Pro) model with a 4k training context length.** Specifically, we divide percentage computational savings into 20 bins $[0\%, 5\%), [5\%, 10\%), \ldots, [95\%, 100\%]$, and for each bin we count the number of heads in the model whose percentage of pruned attention FLOPs and memory accesses falls into that bin. The counts are then normalized to obtain a distribution.

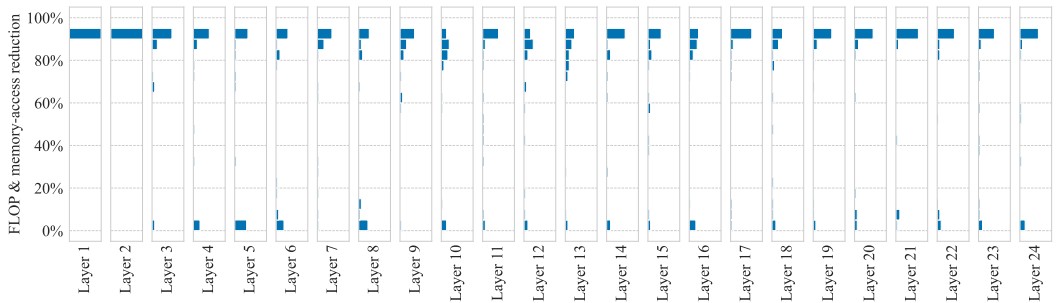

Figure 5: **Distribution of per-head computational savings *in each layer*.** Each column can be seen as a 90°-rotated (and flipped) version of Figure 4, except the distribution is calculated within each layer. The x-axis of each column is the percentage of heads in the corresponding layer whose percentage of pruned FLOPs and memory accesses falls within a specific bin. The range of the x-axis of each column is from 0% to 100%.

## 4.3 Analyses

In this section, we perform a series of analyses to provide deeper insight into our method. First, we show the distribution of computational savings across different attention heads. Second, we visualize the pruning boundaries in some heads. Finally, we investigate how computational savings and model performance vary with $\varepsilon$, the hyperparameter that bounds the total pruned attention weights.

**Distribution of per-head computational savings** In Figure 4, we show the distribution of *per-head* computational savings in a 760M-parameter FoX (Pro) model with a context length of 4k tokens, over the set of all attention heads in the model. Figure 4 shows a clear bimodal pattern, and most attention heads are either "local heads" (most computations are pruned) or "global heads" (only a small proportion or none of the computations are pruned). Furthermore, a majority of the heads are local heads, consistent with the significant FLOP and memory-access savings shown in Figure 2. In Figure 5 we also show the distribution of per-head savings *within each layer*. In general, the distribution for each layer matches the distribution for the entire model, except for the first two layers where all the heads are local.

**Visualization of pruning boundaries** In Figure 6 we show the decay bias matrices $D$ and the attention weight matrices $A$ from three heads in different layers. We also show the

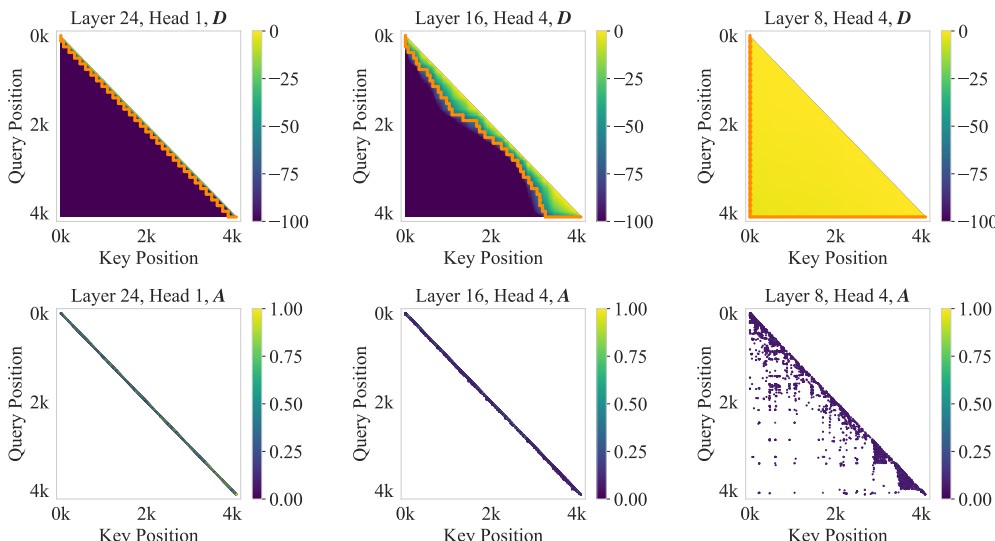

Figure 6: **Visualization of the decay matrices $D$ (top row) and the corresponding attention weight matrices $A$ (bottom row) from three heads in different layers.** The orange line shows the pruning boundary. Since $A$ is very sparse, we only show entries with scores larger than 0.1. These results use a 760M-parameter FoX (Pro) model with a context length of 4k tokens.

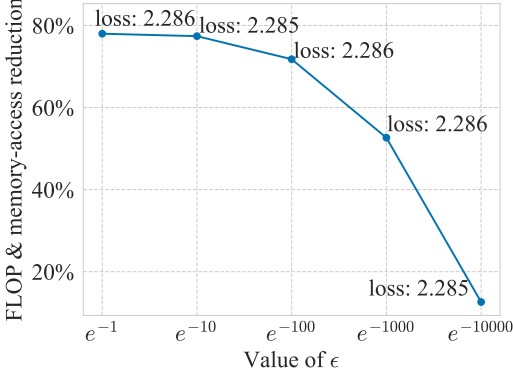

Figure 7: **Impact of $\varepsilon$ on FLOP savings for a 125M-parameter model with a training context length of 16k tokens.** For each data point we also label the corresponding validation loss.

pruning boundaries on the $D$ matrices. The heads on the left and middle are local heads with strong decay, and most off-diagonal blocks are pruned. The rightmost head is a typical global head where no blocks are pruned.

**Effect of varying $\varepsilon$** In Figure 7 we show the impact of $\varepsilon$ – the hyperparameter controlling the maximum total attention weights that can be pruned – on computational savings and language modeling loss. As expected, with smaller $\varepsilon$ the computational savings decrease. On the other hand, there is only marginal gain if one uses a larger $\varepsilon$ (e.g., $e^{-1} \approx 0.37$, which might be unsafe) than our default value $e^{-10} \approx 0.000045$. Therefore, we recommend future work to adopt our default $\varepsilon = e^{-10}$ as it ensures safe pruning while achieving near-optimal computational savings.

## 5 Related work

**Dynamic locality-based computation pruning** The most similar methods to ours are context pruning in Selective Attention (Leviathan et al., 2024) and conditional computation in stick-breaking attention (Tan et al., 2024). Similar to FoX, both Selective Attention and stick-breaking attention learn some forms of data-dependent decay, and thus dynamic pruning similar to our ACP is possible. For Selective Attention, this is done at *inference time* by maintaining a mixed memory budget and dropping KV-cache entries that have the strongest decay. However, it is unclear how this can be adapted for *training*, like what we do in this work with ACP. For stick-breaking attention, this is done by early stopping the stick-breaking process for each query until all attention weights have been assigned. Although for stick-breaking attention conditional computation can also be used in training, Tan et al. (2024) only investigate applying it during inference, so it is unclear how much speed improvement can be obtained when it is applied during training.

**Sliding-window-based computation pruning** Methods such as StreamingLLM (Xiao et al., 2024c), LM-Infinite (Han et al., 2023), MoA (Fu et al., 2024a), and DuoAttention (Xiao et al., 2024b) apply a sliding window mask to pretrained models at inference time to reduce computational costs. This approach is also frequently used in KV-cache eviction methods, and is often combined with some importance-based eviction policy (Zhang et al., 2023; Liu et al., 2023; Ge et al., 2024; Oren et al., 2024; Fu et al., 2024b). With ACP, local heads behave similarly to sliding-window attention. However, unlike these related methods where the window size is typically fixed or based on profiling on some dataset, the "window size" of a local head in ACP is determined by the decay bias matrix and the dynamically set threshold, which guarantees that the total attention weights beyond the local window are negligible.

**Sparse attention** Another category of computation pruning methods exploits the sparsity of softmax attention. These methods mainly differ in how they evaluate the importance of different KV-cache entries based on queries. Most sparse attention methods divide the KV cache into blocks, calculate a summary of each block, and then compute the importance scores using these block summaries (Tang et al., 2024; Xiao et al., 2024a; Gao et al., 2024b; Yuan et al., 2025; Lu et al., 2025). There also exist token-level methods (Desai et al., 2024; Anagnostidis et al., 2023) and more sophisticated methods such as a cluster-based method (Liu et al., 2024) or a mixture of different sparse attention methods (Jiang et al., 2024). These are orthogonal to our locality-based approach, and it is likely that they can be combined with ACP.

## 6 Conclusion

We propose Adaptive Computation Pruning (ACP) for the Forgetting Transformer (FoX), a method that dynamically prunes computations involving input-output dependencies that are strongly decayed by the forget gate in FoX, based on a dynamically set threshold value that ensures negligible impact on the attention output. We apply ACP to language model pretraining and find it leads to significant computational savings and speedups, without sacrificing model performance.

Even though this work primarily focuses on applying ACP during pretraining, we also discuss its potential for inference and present promising preliminary results in the appendix. In particular, for decoding, KV-cache entries could be dynamically evicted based on the pruning boundary, reducing both memory consumption and memory accesses. A more thorough investigation of inference-time ACP is left to future work.

## Acknowledgments

ZL thanks Shawn Tan and Songlin Yang for their helpful discussion. AC acknowledges funding from Microsoft Research. This research was enabled in part by the compute resources, software, and technical help provided by Mila (`mila.quebec`) and the Digital Research Alliance of Canada (`alliance.can.ca`).

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

# A  Proof of upper bound of total pruned attention weights

In this section we prove that when the threshold $\delta$ is properly set, the total pruned attention weights $\sum_{j=1}^{L} \mathbb{1}\{D_{ij} < \delta\} A_{ij}$ would be bounded by a small number $\varepsilon$.

Let $s_{ij} = q_i^{\top} k_j / \sqrt{d}$ and $U$ be an upper bound of $\{|s_{ij}|\}_{i,j \in \{1,...,L\}}$, i.e. $U \geq \max_{i,j \in \{1,...,L\}} |s_{ij}|$. Let $L$ be the sequence length. If we set the threshold to $\delta = -2U - \log L + \log \varepsilon$, then for any $i$ and $j$ such that $D_{ij} < \delta$, we have that (note that $D_{ii} = 0$ by definition):

$$A_{ij} = \frac{\exp(s_{ij} + D_{ij})}{\sum_{k=1}^{i} \exp(s_{ik} + D_{ik})} \leq \frac{\exp(s_{ij} + D_{ij})}{\exp(s_{ii} + D_{ii})} = \exp(s_{ij} - s_{ii} + D_{ij}) \tag{4}$$

$$\leq \exp(|s_{ij} - s_{ii}| + D_{ij}) \leq \exp(2U + D_{ij}) \leq \exp(2U - 2U - \log L + \log \varepsilon) \tag{5}$$

$$= \frac{\varepsilon}{L}. \tag{6}$$

Therefore, we have $\mathbb{1}\{D_{ij} < \delta\} A_{ij} < \frac{\varepsilon}{L}$ for any $i$ and $j$ and $\sum_{j=1}^{L} \mathbb{1}\{D_{ij} < \delta\} A_{ij} < \varepsilon$.

# B  Experimental details

| Configuration | $n_{\text{layers}}$ | $d_{\text{model}}$ | $d_{\text{head}}$ | Peak learning rate |
|---|---|---|---|---|
| 760M params / 16B tokens | 24 | 1536 | 64 | $1 \times 10^{-3}$ |
| 360M params / 7.5B tokens | 24 | 1024 | 64 | $2 \times 10^{-3}$ |
| 125M params / 2.7B tokens | 12 | 768 | 64 | $2 \times 10^{-3}$ |

Table 2: Hyperparameters for different configurations. $n_{\text{layer}}$ counts the number of *blocks*, where each block contains an attention layer and an SwiGLU layer.

Our pretraining hyperparameters follow the setup used in the analysis experiments in Lin et al. (2025). We list the hyperparameters for different training configurations used in this work in Table 2. All models are trained with AdamW (Loshchilov, 2017) with $(\beta_1, \beta_2) = (0.9, 0.95)$, with a linear learning rate warmup from 0 to the peak learning rate for the first $256 \times 2^{20}$ tokens and then a cosine decay to 0. Each training batch contains $0.5 \times 2^{20}$ tokens. All models use a weight decay of 0.1 and gradient clipping of 1.0. We follow the HuggingFace LLaMA initialization and initialize all linear layer weights and embedding parameters with $\mathcal{N}(0, 0.02^2)$. We do not share the parameters between the embedding layer and the output layer. Weight decay is not applied to the RMSNorm parameters and bias terms in linear layers (only the forget gate projection has a bias term). We use `bfloat16` mixed-precision training for all models.

FLOP and memory-access reductions are measured on a 128M-token subset of the LongCrawl64 heldout set. This is calculated as the ratio of the number of pruned blocks to the total number of blocks in the FlashAttention grid. In principle, the FLOP and memory-access savings for the forward pass and the backward pass are different due to different FlashAttention block sizes and thus different grid granularity. However, in practice, the block sizes (smaller than 128) are much smaller than the sequence length (larger than 4k), so the difference is negligible. Therefore, throughout this work, we report the FLOP and memory-access savings for the forward pass.

Attention runtime and training throughput are measured on a 32M-token subset of the same heldout set. The reported attention runtime includes a single forward pass and a single backward pass, measured using `torch.cuda.Events`. When ACP is used, the runtime *includes* the time for boundary index search. When measuring throughput, we use gradient checkpointing and gradient accumulation. Each gradient accumulation step processes 32k tokens. Throughput is measured on 4 NVIDIA L40S GPUs with fully sharded data parallel. The power limit of these GPUs is set to 325W.

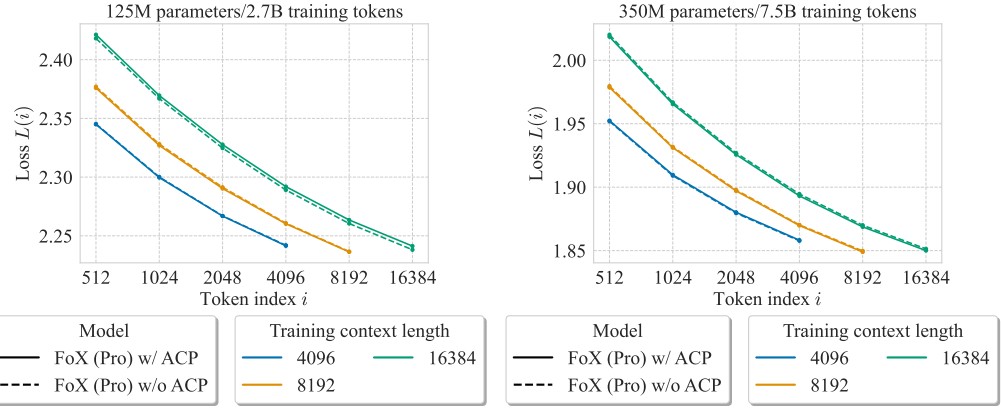

Figure 8: (**left**) Per-token loss given different training context lengths for the 125M-parameter/2.7B-token and 360M-parameter/7.5B-token setting. This is measured on a 2B-token validation set of the LongCrawl64. At each token index $i$, we report the averaged loss over a window of 101 centered at $i$.

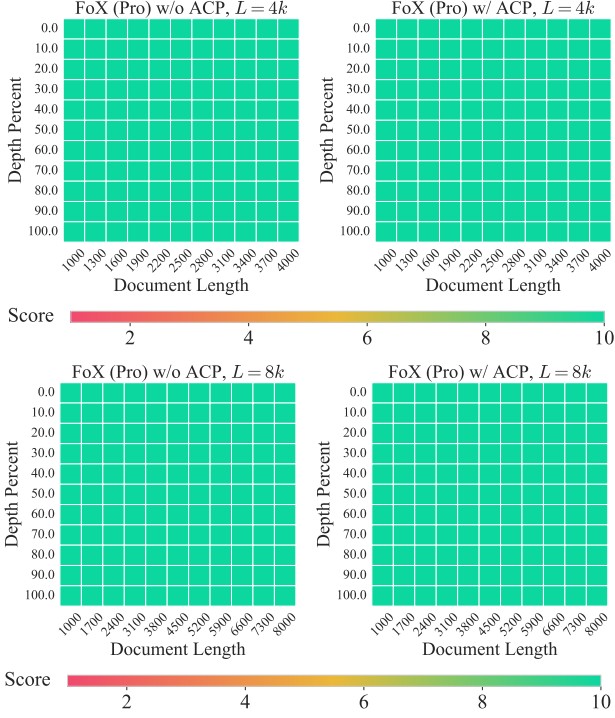

Figure 9: Easy-mode needle-in-a-haystack results for the 760M-parameter models with training context lengths of 4k and 8k tokens.

# C   Additional results

## C.1   Additional FoX (Pro) results

In Figure 8 we show the per-token loss for the 125M-parameter/2.7B-token and 360M-parameter/7.5B-token settings for FoX (Pro) with and without ACP, in addition to the 760M-parameter/16B-token setting in Figure 3 (left). In Figure 9 we show the easy-mode needle-in-a-haystack results for models trained with context lengths of 4k and 8k tokens, respectively, in addition to the 16k-context-length results in Figure 3 (right).

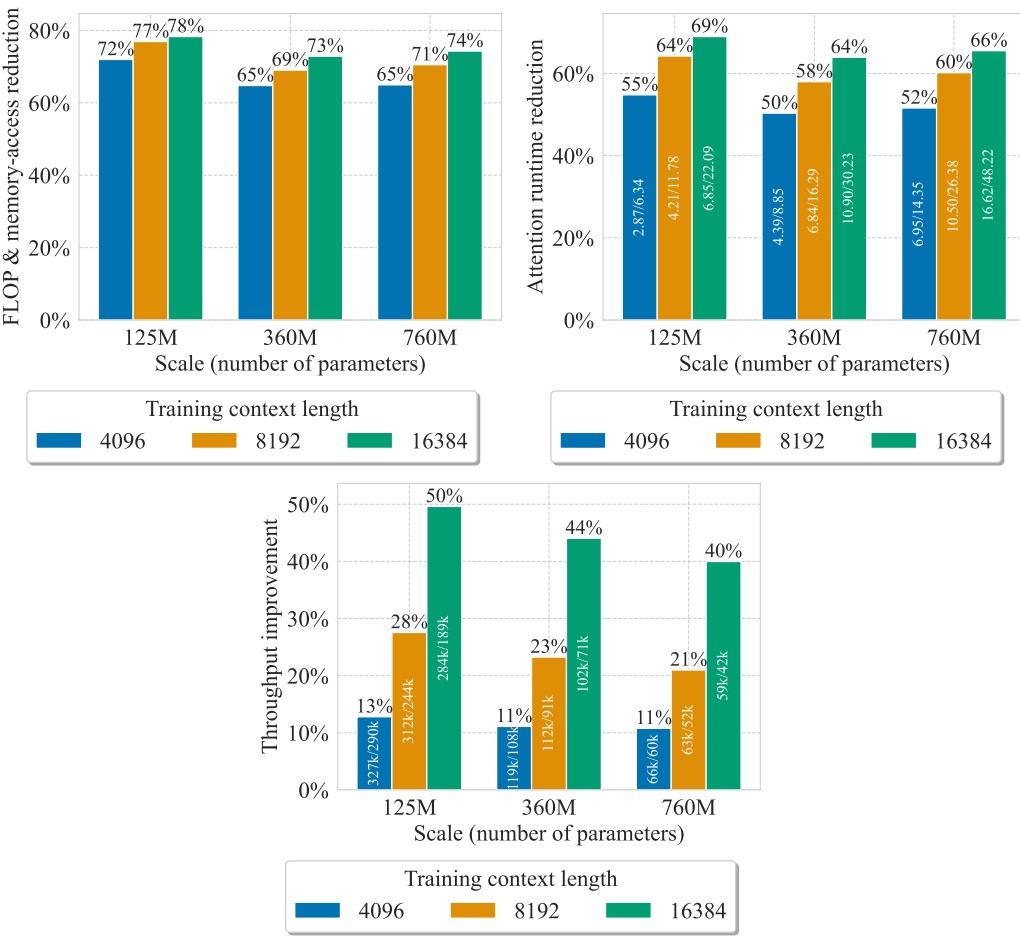

Figure 10: Computational saving and speedup results for FoX (LLaMA). (**left**) Percentage reduction in FLOPs and memory accesses in the attention operation due to ACP. (**right**) Percentage reduction in attention kernel runtime due to ACP. Within each bar we also show the actual runtime with and without ACP in milliseconds. The runtime covers one forward and backward pass on a batch of 0.5M tokens. (**bottom**) Percentage training throughput improvement due to ACP. Within each bar we also show the actual values of training throughput with and without ACP. Throughput is measured in tokens per second. Both the attention kernel runtime and throughput are measured on 4 NVIDIA L40S GPUs.

## C.2   FoX (LLaMA) results

In this section, we present results for the FoX (LLaMA) architecture, in addition to the FoX (Pro) results in the main text.

In Figure 10, we show the percentage reduction in FLOPs and memory accesses *in the attention operation*, the percentage reduction in attention kernel runtime, and the percentage improvement in training throughput due to ACP, across different model sizes and training context lengths, using the FoX (LLaMA) architecture.

In Figure 11 (left) we show the language modeling loss at different token positions for the 760M-parameter FoX (LLaMA) models with different training context lengths, with and without ACP. Figure 11 (right) shows the needle-in-a-haystack retrieval results of the 16k-context-length model in Figure 11 (left), following the "easy mode" setup used in Lin et al. (2025). Table 1 shows the evaluation results on various downstream tasks from Language Model Evaluation Harness (Gao et al., 2024a) for the models in Figure 11 (left).

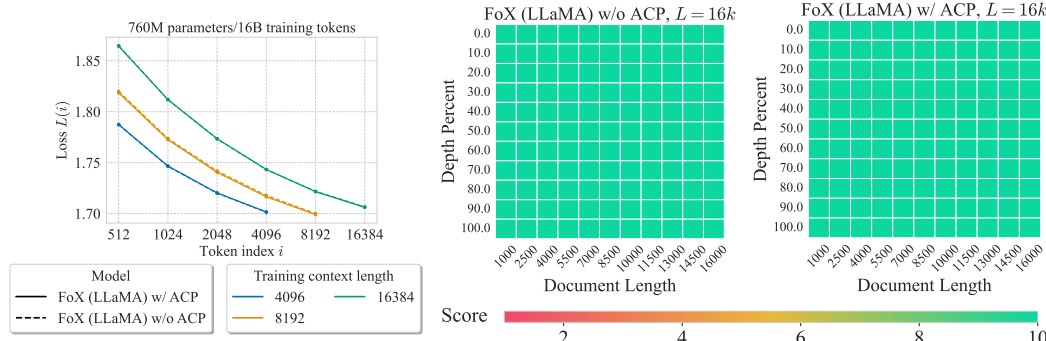

Figure 11: FoX (LLaMA) evlauation results. (**left**) Per-token loss given different training context lengths for the 760M-parameter/16B-token setting. This is measured on a 2B-token validation set of the LongCrawl64. At each token index $i$, we report the averaged loss over a window of 101 centered at $i$. (**right**) Easy-mode needle-in-a-haystack results for the 760M-parameter models with a training context length of $L = 16k$ tokens.

| Model | Wiki. ppl↓ | LMB. ppl↓ | LMB. acc↑ | PIQA acc↑ | Hella. acc-n↑ | Wino. acc↑ | ARC-e acc↑ | ARC-c acc-n↑ | COPA acc↑ | OBQA acc-n↑ | SciQA acc↑ | BoolQ acc↑ | Avg ↑ |
|---|---|---|---|---|---|---|---|---|---|---|---|---|---|
| fot-llama-h64-frac-10-cos-1e-3-end-0-len-4k-760m | 32.90 | 26.42 | 35.36 | 61.37 | 32.46 | 48.78 | 47.73 | 24.66 | 66.00 | 28.20 | 77.50 | 61.22 | 48.33 |
| fot-llama-h64-cos-1e-3-end-0-len-4k-760m | 33.16 | 26.54 | 35.61 | 61.64 | 32.28 | 50.91 | 47.90 | 24.40 | 63.00 | 27.80 | 78.50 | 59.36 | 48.14 |
| fot-llama-h64-frac-10-cos-1e-3-end-0-len-8k-760m | 31.05 | 28.27 | 35.20 | 61.15 | 32.84 | 49.80 | 47.56 | 24.06 | 64.00 | 28.80 | 79.30 | 58.10 | 48.08 |
| fot-llama-h64-cos-1e-3-end-0-len-8k-760m | 31.64 | 27.46 | 36.04 | 62.13 | 32.51 | 50.43 | 48.23 | 23.98 | 65.00 | 27.60 | 78.80 | 56.88 | 48.16 |
| fot-llama-h64-frac-10-cos-1e-3-end-0-len-16k-760m | 31.16 | 28.67 | 35.09 | 61.37 | 31.76 | 50.36 | 47.26 | 24.40 | 66.00 | 28.60 | 77.70 | 60.31 | 48.28 |
| fot-llama-h64-cos-1e-3-end-0-len-16k-760m | 31.03 | 28.41 | 34.89 | 61.21 | 32.27 | 50.51 | 46.68 | 24.06 | 67.00 | 29.60 | 77.30 | 61.07 | 48.46 |

Table 3: **FoX (LLaMA) evaluation results on LM-eval-harness.** All models have roughly 760M non-embedding parameters and are trained on roughly 16B tokens on LongCrawl64. "acc-n" means length-normalized accuracy. $L$ is the training context length.

# D Computational costs of the boundary search algorithm

As discussed in Section 3, the boundary index search algorithm has a linear complexity of $O(\max(\frac{L}{B_q}, \frac{L}{B_k}))$, compared to the $O(L^2 d)$ quadratic complexity of standard full attention. Note that even though this algorithm runs sequentially, in practice it still has negligible *wall-clock time* compared to actual attention computations, mainly because FlashAttention also runs sequentially within each thread block with a similar number of iteration steps as this algorithm.

In Table D we report the percentage of wall clock time spent on boundary index search within the attention kernel, i.e., boundary search time divided by the total runtime of the attention kernel. Note the total runtime of the attention kernel *includes the time for boundary index search*. As shown in this table, the computational costs of boundary index search are minimal.

|  | 125M | 350M | 760M |
|---|---|---|---|
| $L = 4096$ | 4% | 3% | 2% |
| $L = 8192$ | 5% | 4% | 3% |
| $L = 16384$ | 6% | 4% | 3% |

Table 4: Percentage of wall-clock time spent on boundary index search within the attention kernel given different model sizes and sequence lengths $L$.

# E Obtaining an attention logit upper bound from QK-norm parameters

When QK-norm is used (assuming we use RMSNorm (Zhang & Sennrich, 2019), as in the FoX (Pro) architecture), the L2-norms of queries and keys are bounded by $\gamma^k \sqrt{d}$ and $\gamma^q \sqrt{d}$ respectively, where $\gamma^k = \max_{i \in \{1,...,d\}} |\gamma_i^k|$ is the maximum magnitude of the key RMSNorm scaling parameters $\{\gamma_i^k\}_{i=1}^d$ and $\gamma^q$ is defined similarly. Therefore $|s_{ij}| \leq \frac{\|q_i\|_2 \|k_j\|_2}{\sqrt{d}} \leq \gamma^k \gamma^q \sqrt{d}$ and thus we can set $U = \gamma^k \gamma^q \sqrt{d}$.

# F Inference-time ACP

In this section we discuss how ACP can be used at inference time and present some preliminary results. Applying ACP to prefilling is straightforward, so we mainly discuss ACP for decoding.

For decoding, due to the monotonicity of the pruning boundary, we could maintain a pruning boundary index $j$ for each head and update it in an online fashion. Specifically, whenever $D_{ij} < \delta$ – where $i$ is the current timestep – we increment $j$ until $D_{ij} \geq \delta$. Since $j$ never decreases, we can discard any KV-cache entries beyond the pruning boundary, thus reducing memory consumption and memory accesses.

In our preliminary results with a naive implementation that does not perform explicit KV cache eviction (but still skips loading pruned blocks to shared memory), applying ACP during inference achieves the same level (around 70%) reduction in memory accesses and FLOPs. This is expected as these savings are implementation-agnostic. However, our analysis shows that our implementation is likely bottlenecked by the kernel launch overheads during decoding, and wall-clock time improvement is most obvious with long prefilling lengths and large batch sizes. Specifically, with a sufficiently large prefilling length and batch size, ACP reduces the per-step attention kernel runtime by 50% to 60%, which is similar to the 50% to 70% reduction we see during pretraining. However, even in this setup, we do not see a clear improvement in end-to-end decoding throughput. Our analysis shows this is likely because the kernel execution overlaps with the significant Triton kernel launch overheads (many components in our implementation such as RMSNorm are in Triton). Therefore, reducing the attention kernel runtime has little effect on end-to-end throughput, as the throughput would still be bottlenecked by these kernel launch overheads.

We comment that kernel launch overheads could be largely mitigated with optimization (e.g., CUDA graph), so we expect that with a properly optimized implementation, ACP should be able to bring the same level of speedup to LLM decoding as it does to pretraining, especially given that the percentage reductions in memory accesses brought by ACP are similar during pretraining and decoding.

