# OpenReview forum: "Adaptive Computation Pruning for the Forgetting Transformer"
_colmweb.org/COLM/2025/Conference — COLM 2025_

### Official Review · Reviewer_s3o6 · 2025-05-09

**Rating:** 6
**Confidence:** 5
**Ethics Flag:** 1

**Summary:**

This paper observes that many attention heads in the FoX (Forgetting Transformer) architecture tend to forget information rapidly, resulting in outputs that primarily depend on local context at each timestep. Motivated by this, the authors propose Adaptive Computation Pruning (ACP), a method that dynamically prunes input-output dependencies in attention computation when they are significantly suppressed by the forget gate. This is done using a dynamic pruning threshold to ensure that the removed attention weights remain negligible.

ACP is applied to FoX-based language model pretraining and demonstrates consistent improvements: it reduces softmax attention FLOPs by over 70% across varying model sizes and context lengths, leading to a 10%–40% increase in training throughput. The benefit becomes more pronounced as the context length increases.

**Reasons To Accept:**

1. The paper is clearly written and well-organized.
2. The idea of adaptive pruning for *Forgetting Transformer* is interesting and practically motivated.

**Reasons To Reject:**

1. The proposed method essentially combines the Forgetting Transformer's attention mechanism with an adaptive sliding-window strategy. This idea is fairly common and lacks strong novelty.
2. The technique appears to be generally applicable to standard multi-head self-attention as well, not necessarily tied to the Forgetting Transformer. It is unclear whether the authors have conducted any experiments to explore this applicability.
3. The computational overhead of the method is not thoroughly discussed. Since pruning is performed at the block level and requires determining block-wise indices, it may not be computationally lightweight. Moreover, it does not appear to guarantee globally optimal pruning, raising concerns about its efficiency compared to other pruning strategies.
4. The experimental section lacks comparative baselines. The results only compare the proposed method to unpruned FoX, making it difficult to assess the overall competitiveness or practical advantage of ACP. It's possible that its performance remains below that of existing pruning methods.
5. Some reported results in this paper appear questionable. The FoX (Pro) result on HellaSwag is around 33 in this paper, while the original FoX paper reports a score of 38.39. It's unclear why a different dataset size was used. Furthermore, the original FoX paper provided evaluation on LongBench, which is not included here, missing the way for a more comprehensive and fair comparison.

---

> ### Author Response · Authors · 2025-06-03
> **Author Response**
>
> Thanks you for your review and valuable feedback!
>
> First, we invite you to read our [General Response](https://openreview.net/forum?id=xNj14CY5S1&noteId=DHRZyGenS8) to all reviewers, which includes additional results measuring the speed-up of the attention kernel in terms of wall-clock time. Specifically, we demonstrate a 50% to 70% reduction in attention runtime—equivalent to a two- to three-fold speed-up.
>
> > The proposed method essentially combines the Forgetting Transformer's attention mechanism with an adaptive sliding-window strategy. This idea is fairly common and lacks strong novelty.
>
>
> ACP is specifically designed for FoX and leverages the unique properties of its decay matrix to enable safe and efficient pruning. For instance, identifying a pruning boundary in linear time with respect to $L$ is made possible by the monotonicity of $D_{ij}$ with respect to both $i$ and $j$—a naive approach would require examining all $L^2$ entries of $D_{ij}$. **The sliding-window-like pruning pattern also naturally emerges from the monotonicity of $D_{ij}$**, rather than being manually engineered into the method.
>
> Even the basic feasibility of pruning depends on recognizing two key facts: (1) for finite-length sequences, attention logits are bounded and can be dominated by $D_{ij}$, making safe pruning with theoretical guarantees possible; and (2) most attention heads are local, so pruning can yield substantial efficiency gains. These insights, and the algorithms derived from them, are both novel and specifically tailored for FoX—a recent innovation in itself.
>
> In contrast to our approach, which is both virtually lossless and efficient, most sliding-window attention-based pruning methods either impose a fixed window size or determine the window size through profiling on certain datasets. These strategies typically lack the strong theoretical guarantees that our method provides. More importantly, the majority of these approaches are designed solely for inference with pretrained models and cannot be applied during pretraining, making them fundamentally incomparable to our work. As discussed in our related work section, there are also approaches that employ alternative attention mechanisms—such as stick-breaking attention—for pruning. These non-softmax-based methods are orthogonal to our work.
>
>
> > The technique appears to be generally applicable to standard multi-head self-attention as well, not necessarily tied to the Forgetting Transformer. It is unclear whether the authors have conducted any experiments to explore this applicability.
>
> As mentioned earlier, our approach fundamentally relies on the decay bias matrix introduced in FoX and several of its properties to enable pruning. Without $D_{ij}$, there would be no special structure to exploit for safe and efficient pruning. Notably, pruning based on a threshold applied directly to the attention logits $q_i^\top k_j$—rather than to $D_{ij}$—would require computing all logits, which incurs a complexity of $O(L^2 d)$ (essentially full attention) and is therefore impractical. In contrast, the ACP boundary search algorithm operates with linear complexity in the sequence length and incurs negligible runtime overhead in practice.
>
> > The computational overhead of the method is not thoroughly discussed. Since pruning is performed at the block level and requires determining block-wise indices, it may not be computationally lightweight.
>
>
> As noted in the General Response, the complexity of the boundary index search is $O(\max(M, N))$, where $M = L / B_q$ and $N = L / B_k$. In other words, the complexity is linear with respect to the sequence length $L$ (note that in Algorithm 1, we never reset $l$ to 1 within the for loop). This cost is negligible compared to the quadratic $O(L^2 d)$ complexity of the actual attention computation.
>
> Below, we report the fraction of wall-clock time spent on the boundary index search within the attention kernel for various context lengths and model sizes (i.e., the boundary index search time divided by the total runtime of the attention kernel).
>
>
> |             | 125M | 360M | 760m |
> | :---------- | :--- | :--- | :--- |
> | $L = 4096$  | 4%   | 3%   | 2%   |
> | $L = 8192$  | 5%   | 4%   | 3%   |
> | $L = 16384$ | 6%   | 4%   | 3%   |
>
>
>
> As shown in the Table, the time spent on boundary index search is negligible (always below 6%).
>
> > Moreover, it does not appear to guarantee globally optimal pruning, raising concerns about its efficiency compared to other pruning strategies.
>
> ACP already achieves provably near-lossless pruning while delivering significant speedups. We are unclear on what is meant by "globally optimal pruning" in this context. Could you clarify what specific criteria you are referring to?

---

> > ### Author Response · Authors · 2025-06-03
> > **Author Response (cont.)**
> >
> > > The experimental section lacks comparative baselines. The results only compare the proposed method to unpruned FoX, making it difficult to assess the overall competitiveness or practical advantage of ACP. It's possible that its performance remains below that of existing pruning methods.
> >
> > Our method is designed specifically for FoX so it is natural to measure how much it speeds up (unpruned) FoX. As mentioned before, most "pruning" methods either deviate from standard softmax attention or are only designed for inference of already pretrained models, so they are not comparable to our method which is applicable during pretraining. Furthermore, our method is guaranteed to be virtually lossless (note our default $\varepsilon=e^{-10}\approx 0.000045$ is even below the roughly $0.001$ to $0.01$ relative error introduced by the BF16 mixed training) and only requires minimal modifications to FlashAttention, so in principle it could be combined with any other (lossy) pruning methods. In this sense, ACP can be seen as orthogonal to all other pruning methods.
> >
> > > Some reported results in this paper appear questionable. The FoX (Pro) result on HellaSwag is around 33 in this paper, while the original FoX paper reports a score of 38.39. It's unclear why a different dataset size was used.
> >
> > As you mentioned, the results are different because we use 16B tokens for the 760M-parameter models, while the main results in the FoX paper use 48B tokens. This choice is mainly due to the large number of experiments we need to run (combinations of different model sizes, context lengths, with and without ACP) and our limited computational resources. Also note that our set of configurations (125M-param/2.7B-token, 360M-param/7.5B-token, 760M-param/16B-token) are from the ablation experiments in the FoX paper which aligns with the Chinchilla scaling law.
> >
> > For completeness, we have also trained a 760M FoX model for 48B tokens with ACP. We compare the results with the official results as follows. In the following results marked "FoX (Pro) w/o ACP" are either taken from the FoX paper or are reproduced using their official checkpoint.
> >
> > **Per-token loss at different token positions**:
> >
> >
> > |                   | 512   | 1024  | 2048  | 4096  | 8192  | 16384 |
> > | :---------------- | :---- | :---- | :---- | :---- | :---- | :---- |
> > | FoX (Pro) w/ ACP  | 1.635 | 1.586 | 1.550 | 1.523 | 1.502 | 1.487 |
> > | FoX (Pro) w/o ACP | 1.635 | 1.584 | 1.549 | 1.521 | 1.501 | 1.486 |
> >
> > **Language Model Evaluation Harness**:
> >
> > |                                        | Wiki.           | LMB.            | LMB.          | PIQA          | Hella.          | Wino.         | ARC-e         | ARC-c           | COPA          | OBQA            | SciQA         | BoolQ         | Acc Avg    |
> > | :------------------------------------- | :-------------- | :-------------- | :------------ | :------------ | :-------------- | :------------ | :------------ | :-------------- | :------------ | :-------------- | :------------ | :------------ | :--------- |
> > |                                        | ppl$\downarrow$ | ppl$\downarrow$ | acc$\uparrow$ | acc$\uparrow$ | acc-n$\uparrow$ | acc$\uparrow$ | acc$\uparrow$ | acc-n$\uparrow$ | acc$\uparrow$ | acc-n$\uparrow$ | acc$\uparrow$ | acc$\uparrow$ | $\uparrow$ |
> > | FoX (Pro) w/o ACP, $L=16k$, 48B tokens | 23.04           | 14.91           | 42.75         | 64.09         | 38.39           | 52.33         | 52.23         | 26.54           | 71.00         | 29.80           | 85.10         | 46.57         | 50.88      |
> > | FoX (Pro) w/ ACP, $L=16k$, 48B tokens  | 23.44           | 14.46           | 43.99         | 64.31         | 37.70           | 52.57         | 51.22         | 24.32           | 64.00         | 32.20           | 84.40         | 57.65         | 51.24      |
> >
> > **LongBench**
> >
> >
> > |                                        | NarrativeQA | Qasper | MFQA  | HotpotQA | 2WikiMQA | Musique | GovReport | QMSum | MultiNews | TREC  | TriviaQA | SamSum | LCC   | RepoBench-P |
> > | :------------------------------------- | :---------- | :----- | :---- | :------- | :------- | :------ | :-------- | :---- | :-------- | :---- | :------- | :----- | :---- | :---------- |
> > | FoX (Pro) w/o ACP, $L=16k$, 48B tokens | 13.38       | 18.88  | 28.73 | 15.27    | 25.39    | 6.49    | 22.71     | 13.51 | 12.27     | 63.50 | 37.36    | 22.74  | 10.90 | 9.10        |
> > | FoX (Pro) w/ ACP, $L=16k$, 48B tokens  | 13.15       | 16.24  | 27.26 | 15.14    | 24.72    | 6.46    | 24.49     | 13.44 | 6.73      | 62.00 | 42.34    | 20.41  | 9.37  | 14.12       |
> >
> >
> > As shown in these results, the per-token language modeling loss almost match exactly and the downstream task results are similar. As mentioned in our paper and also will be discussed later, downstream task performance can vary greatly across training runs so it is not possible to get exactly the same results. We also run the needle-in-a-haystack test and FoX with ACP achieves perfect accuracy, similar to FoX without ACP.

---

> > ### Author Response · Authors · 2025-06-03
> > **Author Response (cont.)**
> >
> > Below we show FLOP savings, attention runtime reduction, training throughput improvement provided by ACP for FoX trained with 48B tokens. We also show results for models trained with 16B tokens for comparison. The results are as follows:
> >
> > |                      | FLOP savings | Attention Runtime Reduction | Throughput Improvment |
> > | :------------------- | :----------- | :-------------------------- | :-------------------- |
> > | FoX (Pro) 16B tokens | 78%          | 70%                         | 43%                   |
> > | FoX (Pro) 48B tokens | 70%          | 63%                         | 35%                   |
> >
> > As shown in the results, the improvements for 48B-token models are slightly less but still signficant.
> >
> > > Furthermore, the original FoX paper provided evaluation on LongBench, which is not included here, missing the way for a more comprehensive and fair comparison.
> >
> > LongBench is a generation-based benchmark for instruction-tuned models, so the results can have high variance for small base models that have not been instruction-tuned. Therefore, it is less suitable for our purposes. To demonstrate this, we have repeated three runs for the 760M-param/16B-token/4k-context length setting with unpruned FoX:
> >
> >
> > |                                   | NarrativeQA | Qasper | MFQA  | HotpotQA | 2WikiMQA | Musique | GovReport | QMSum | MultiNews | TREC  | TriviaQA | SamSum | LCC   | RepoBench-P |
> > | :-------------------------------- | :---------- | :----- | :---- | :------- | :------- | :------ | :-------- | :---- | :-------- | :---- | :------- | :----- | :---- | :---------- |
> > | FoX (Pro) w/o ACP, $L=4k$, seed 0 | 9.94        | 16.01  | 18.71 | 8.22     | 17.42    | 3.94    | 22.97     | 11.69 | 14.42     | 41.00 | 21.42    | 13.42  | 8.61  | 10.67       |
> > | FoX (Pro) w/o ACP, $L=4k$, seed 1 | 9.06        | 10.74  | 19.27 | 9.84     | 18.84    | 3.87    | 20.74     | 11.60 | 12.80     | 46.00 | 21.15    | 5.10   | 11.01 | 9.34        |
> > | FoX (Pro) w/o ACP, $L=4k$, seed 2 | 10.85       | 15.01  | 19.70 | 7.13     | 13.68    | 4.95    | 21.66     | 11.03 | 13.85     | 43.50 | 20.80    | 12.40  | 7.11  | 11.14       |
> >
> > Note the large variance across seeds especially for some tasks such as Qasper and Samsum.
> >
> > For completeness, we have run LongBench evaluation for the 760M-param models trained with 16B tokens with different context lengths. The results are follows:
> >
> >
> > |                            | NarrativeQA | Qasper | MFQA  | HotpotQA | 2WikiMQA | Musique | GovReport | QMSum | MultiNews | TREC  | TriviaQA | SamSum | LCC  | RepoBench-P |
> > | :------------------------- | :---------- | :----- | :---- | :------- | :------- | :------ | :-------- | :---- | :-------- | :---- | :------- | :----- | :--- | :---------- |
> > | FoX (Pro) w/ ACP, $L=4k$   | 8.63        | 14.53  | 19.12 | 9.87     | 20.77    | 4.27    | 19.81     | 9.24  | 13.20     | 38.50 | 20.15    | 7.52   | 5.77 | 14.74       |
> > | FoX (Pro) w/o ACP, $L=4k$  | 9.94        | 16.01  | 18.71 | 8.22     | 17.42    | 3.94    | 22.97     | 11.69 | 14.42     | 41.00 | 21.42    | 13.42  | 8.61 | 10.67       |
> > | FoX (Pro) w/ ACP, $L=8k$   | 9.63        | 13.58  | 20.61 | 9.06     | 19.18    | 4.15    | 20.48     | 12.78 | 4.04      | 56.50 | 23.38    | 14.27  | 9.49 | 7.50        |
> > | FoX (Pro) w/o ACP, $L=8k$  | 10.79       | 11.52  | 21.23 | 8.32     | 19.78    | 4.67    | 20.93     | 11.49 | 12.76     | 56.50 | 20.08    | 10.35  | 7.01 | 7.87        |
> > | FoX (Pro) w/ ACP, $L=16k$  | 9.40        | 15.96  | 20.64 | 8.47     | 10.80    | 3.75    | 21.90     | 8.78  | 7.41      | 62.50 | 21.30    | 18.47  | 6.02 | 3.07        |
> > | FoX (Pro) w/o ACP, $L=16k$ | 9.19        | 11.34  | 20.32 | 10.03    | 18.77    | 5.05    | 22.16     | 11.46 | 10.80     | 40.00 | 26.13    | 18.88  | 8.17 | 4.52        |
> >
> > As shown in the table, the results for models trained with 16B tokens with and without ACP remain similar, with some variations in some tasks. As discussed above, this is likely due to variance across training runs. Note that LongBench results for models trained with 48B tokens are also included in the response to your last concern regarding dataset size.
> >
> > We hope our response addresses your concerns. If so, we would greatly appreciate it if you could raise your score. Feel free to let us know if you have more questions!

---

> > > ### Comment · Reviewer_s3o6 · 2025-06-06
> > >
> > > The authors have provided very detailed explanations and experimental results in the rebuttal to address my concerns, particularly clarifying the discrepancies with the original FoX paper. I appreciate their efforts and will raise my rating accordingly.

---

> > > > ### Author Response · Authors · 2025-06-06
> > > >
> > > > Thank you for raising your score! We're glad our clarification helped.

---

### Official Review · Reviewer_tEnv · 2025-05-13

**Rating:** 7
**Confidence:** 2
**Ethics Flag:** 1

**Summary:**

Adaptive Computation Pruning (ACP) introduces a block‐wise attention pruning mechanism that dynamically discards negligible attention scores during training to reduce computation without sacrificing model quality. By partitioning the attention matrix into fixed-size blocks, ACP computes for each block a tight bound on its maximum contribution (using precomputed norms U) and derives a threshold δ that guarantees any value below it has a provably negligible impact on the softmax output. During each forward pass, a simple boundary‐search algorithm identifies and skips entire blocks whose scores fall below δ, yielding substantial speedups. ACP integrates seamlessly into FoX‐style transformer pretraining, provides theoretical guarantees on output fidelity, and demonstrates significant efficiency gains with minimal accuracy loss across standard benchmarks.

**Questions To Authors:**

Could you discuss how the choice of block sizes (Bq, Bk) and the resulting grid dimensions (M, N) influence the trade-off between pruning granularity and the overhead associated with boundary detection?

**Reasons To Accept:**

- Clearly motivated problem with provable guarantees.

**Reasons To Reject:**

Although the authors claim that identifying boundary indices is “negligible” compared to attention computation, there is no quantitative complexity analysis or profiling to substantiate this .

- The threshold \delta = –2U – log L + log \epsilon seems to derive directly from bounding softmax tails by exp(2U + Dij)≤\epsilon/L, a technique common in sparse-attention analyses.

- Although the authors generalize the method to block-level algorithms, the approach remains a straightforward application of existing techniques.

- While Fig 7 shows pruning percentages for varying \epsilon, there is no guidance on how to choose \epsilon in novel tasks or architectures, leaving unclear about trade-offs between speed and accuracy .

- There is no discussion of how choices of block sizes (Bq, Bk) and resulting grid dimensions (M, N) affect the granularity of pruning versus the overhead of boundary detection .

---

> ### Author Response · Authors · 2025-06-03
> **Author Response**
>
> Thanks you for your review and valuable feedback!
>
> First, we invite you to read our [General Response](https://openreview.net/forum?id=xNj14CY5S1&noteId=DHRZyGenS8) to all reviewers, which includes additional results measuring the speed-up of the attention kernel in terms of wall-clock time. Specifically, we demonstrate a 50% to 70% reduction in attention runtime—equivalent to a two- to three-fold speed-up.
>
>
> We address your specific concerns as follows:
>
> > Although the authors claim that identifying boundary indices is “negligible” compared to attention computation, there is no quantitative complexity analysis or profiling to substantiate this .
>
> Below we report the fraction of wall clock time spent on the boundary index search algorithm within the attention kernel for different context lengths and model sizes (i.e., boundary index search time divided by the total runtime of the attention kernel)
>
>
> |             | 125M | 360M | 760m |
> | :---------- | :--- | :--- | :--- |
> | $L = 4096$  | 4%   | 3%   | 2%   |
> | $L = 8192$  | 5%   | 4%   | 3%   |
> | $L = 16384$ | 6%   | 4%   | 3%   |
>
> As shown in the table, the time spent on boundary index search is always below 6%. As mentioned in the General Response, the complexity of boundary index search is $O(\max(M, N))$, where $M = L/B_q$ and $N=L/B_k$. In other words, the complexity is **linear with respect to the sequence length $L$** (note in Algorithm 1 we never reset $l$ to 1 within the for loop). This is negligible compared to the quadratic $O(L^2 d)$ cost of the actual attention computation.
>
> > The threshold \delta = –2U – log L + log \epsilon seems to derive directly from bounding softmax tails by exp(2U + Dij)≤\epsilon/L, a technique common in sparse-attention analyses.
>
> Yes, this is derived as you described (also detailed in Appendix A). Though this is a straightforward result, we believe it is specific to FoX as it involves the FoX decay bias which is not present in standard softmax attention.
>
>
> > Although the authors generalize the method to block-level algorithms, the approach remains a straightforward application of existing techniques.
>
> ACP is specifically designed for FoX and leverages the unique properties of its decay matrix to enable safe and efficient pruning. For instance, identifying a pruning boundary in linear time with respect to $L$ is made possible by the monotonicity of $D_{ij}$ with respect to both $i$ and $j$—a naive approach would require examining all $L^2$ entries of $D_{ij}$. The sliding-window-like pruning pattern also naturally emerges from the monotonicity of $D_{ij}$, rather than being manually engineered into the method. Even the basic feasibility of pruning depends on recognizing two key facts: (1) for finite-length sequences, attention logits are bounded and can be dominated by $D_{ij}$, making safe pruning with theoretical guarantees possible; and (2) most attention heads are local, so pruning can yield substantial efficiency gains. These insights, and the algorithms derived from them, are both novel and specifically tailored for FoX—a recent innovation in itself. Thus we believe ACP is far from a "straightforward application of existing techniques."
>
> > While Fig 7 shows pruning percentages for varying \epsilon, there is no guidance on how to choose \epsilon in novel tasks or architectures, leaving unclear about trade-offs between speed and accuracy .
>
> Our default $\varepsilon=e^{-10}\approx 0.000045$ already guarantees virtually lossless pruning and this is our recommendation for future work (note this is even below the roughly $0.001$ to $0.01$ relative error introduced by BF16 mixed precision training). In addition, Fig 7 shows there would not be much speed gain if one uses a larger  $\varepsilon$ (e.g., $e^{-1}$, which might be unsafe) than our default value. This means one can simply set $\varepsilon =e^{-10}$ and get virtually lossless pruning and near optimal speed without any trade-off.

---

> > ### Author Response · Authors · 2025-06-03
> > **Author Response (cont.)**
> >
> > > There is no discussion of how choices of block sizes (Bq, Bk) and resulting grid dimensions (M, N) affect the granularity of pruning versus the overhead of boundary detection .
> >
> > Typical values of $B_q$ and $B_k$ are 32, 64, 128 (chosen based on profiling), which are limited by GPU's on-chip shared memory size. This is very small compared to typical sequence lengths such as 4k and 16k, so their impact on pruning granularity is negligible. As mentioned before the complexity of boundary index search is $O(\max(M, N))$ where $M = L/B_q$ and $N=L/B_k$, which is negligible compared to the $O(Ld^2)$ complexity the attention computation regardless $B_q$ and $B_k$. In the following tables we show the FLOP savings (which depends on pruning granularity) and the fraction of runtime spent on the boundary index search algorithm within the attention kernel for different ($B_q$, $B_k$), for a 760M-parameter model trained with a context length of 4k tokens. Note that we separate the forward pass and backward pass because they use different block sizes.
> >
> > Forward pass results are shown as follows:
> >
> >
> > |                                                 | (128, 64) (default) | (64, 64) | (32, 32) |
> > | :---------------------------------------------- | :------------------ | :------- | :------- |
> > | Fraction of time spent on boundary index search | 3%                  | 3%       | 4%       |
> > | FLOP savings                                    | 72%                 | 73%      | 74%      |
> >
> > Backward pass results are shown as follows. In our implementation the gradient computation for queries and keys/values use different kernels and have different block sizes, so there are two pairs of $(B_q, B_k)$  in each column (one for the query grad kernel and one for key/value grad kernel).
> >
> > |                                                 | ((128, 32), (64, 128)) (default) | ((64, 64), (64, 64)) | ((32, 32), (32, 32)) |
> > | :---------------------------------------------- | :------------------------------- | :------------------- | :------------------- |
> > | Fraction of time spent on boundary index search | 2%                               | 2%                   | 3%                   |
> > | FLOP savings                                    | 72%                              | 73%                  | 74%                  |
> >
> >  As can be seen in these results, the block sizes have little impact on boundary search overheads and FLOP savings.
> >
> > We hope our response addresses your concerns. If so, we would greatly appreciate it if you could raise your score. Feel free to let us know if you have more questions!

---

> > > ### Comment · Reviewer_tEnv · 2025-06-05
> > >
> > > Thanks for your response. I appreciate the additional details. After going through the response, I have raised my score.

---

> > > > ### Author Response · Authors · 2025-06-05
> > > >
> > > > Thank you for raising your score! We’re glad the additional details helped.

---

### Official Review · Reviewer_KGU7 · 2025-05-14

**Rating:** 6
**Confidence:** 4
**Ethics Flag:** 1

**Summary:**

The paper introduce Adaptive Computation Pruning (ACP) for the Forgetting Transformer, using the decay bias D_ij to skip entire query–key blocks whose maximum value falls below a theoretically derived threshold delta = −2U − log L + log epsilon, which guarantees that no more than an epsilon fraction of attention mass is discarded.

ACP is implemented as a two-stage routine that first finds a monotone block boundary in O(MN) time and then runs FlashAttention only on the retained region; experiments on 125 M–760 M-parameter models and context lengths up to 16 k show 70–79 percent attention-FLOP savings and 10–40 percent wall-clock speed-ups without much loss on language-modeling and retrieval benchmarks.

**Questions To Authors:**

Can you quantify end-to-end speed-ups and memory savings during pre-fill and autoregressive decoding when ACP is applied to KV-cache eviction, and how do these gains compare with the training-time numbers?

What fraction of total attention runtime is spent on the boundary-search algorithm at sequence length 16 k on L40S versus A100 GPUs

**Reasons To Accept:**

The pruning threshold is mathematically solid, with a compact proof that bounds the discarded attention mass and thus preserves model fidelity; ACP operates at the FlashAttention block granularity, aligning with GPU memory-coalescing patterns and requiring only minimal Triton changes for deployment. Experiments demonstrate large training-time throughput gains without accuracy loss and provide insightful visual diagnostics that explain why certain head become almost completely local.

**Reasons To Reject:**

The method is tightly coupled to the FoX decay bias and its generalizability to standard RoPE transformers or other positional encodings is not very clear; all reported results concern training, leaving potential inference-time benefits during autoregressive decoding—where KV-cache bandwidth dominates. scalability is only tested up to 760 M parameters on a single 4 × L40S node, so behavior on multi-billion-parameter models and distributed settings remains unknown.

---

> ### Author Response · Authors · 2025-06-03
> **Author Response**
>
> Thank you for your review and valuable feedback!
>
> First, we invite you to read our [General Response](https://openreview.net/forum?id=xNj14CY5S1&noteId=DHRZyGenS8) to all reviewers, which includes additional results measuring the speed-up of the attention kernel in terms of wall-clock time. Specifically, we demonstrate a 50% to 70% reduction in attention runtime—equivalent to a two- to three-fold speed-up.
>
> We address your specific concerns below:
>
> > The method is tightly coupled to the FoX decay bias and its generalizability to standard RoPE transformers or other positional encodings is not very clear;
>
> Indeed our method relies on the special properties of the FoX decay bias matrix (in particular the monotonicity property) for efficient and safe pruning. However, note that it is possible to add RoPE to FoX or add the FoX decay bias to standard RoPE-based Transformers. Below we show the FLOP savings, attention runtime reduction, and training throughput improvements when ACP is applied to FoX + RoPE for a 125M-param model with a context length of 16k. We also show results for FoX without RoPE for comparison
>
> |                     | FLOP savings | Attention Runtime Reduction | Throughput Improvment |
> | :------------------ | :----------- | :-------------------------- | :-------------------- |
> | FoX (Pro) w/o RoPE  | 77%          | 68%                         | 45%                   |
> | FoX (Pro) with RoPE | 73%          | 64%                         | 41%                   |
>
> As shown in the table, the speedup for FoX + RoPE is slightly less than that for FoX without RoPE, but still significant.
>
> We also make several comments:
>
> 1. The FoX paper shows that RoPE does not improve performance for FoX. This is also the case in our experiments.
> 2. The FoX paper shows FoX without RoPE generally outperforms standard RoPE Transformers, especially for long context pretraining. So it is possible that future models could simply replace RoPE with the FoX decay bias.
> 3. It is possible to the add FoX decay bias to pretrained RoPE-based Transformers (e.g., LLaMA and Qwen) and finetune them, after which ACP could be applied.

---

> > ### Author Response · Authors · 2025-06-03
> > **Author Response (cont.)**
> >
> > > Can you quantify end-to-end speed-ups and memory savings during pre-fill and autoregressive decoding when ACP is applied to KV-cache eviction, and how do these gains compare with the training-time numbers?
> >
> > In our implementation, prefilling is a simple forward pass, so the improvements are comparable to those for pretraining (e.g., 34% improvement in end-to-end prefilling speed compared to the 43% pretraining throughput improvement).
> >
> > For decoding, note the percentage of memory IO savings provided by ACP is the same as the percentage of FLOP savings because the kernel does not need to load pruned KV blocks. In our preliminary results with our simple FlashAttention decoding implementation, we observe that memory IO savings and attention kernel runtime during decoding are comparable to those during pretraining. However, **we do not observe a clear improvement in end-to-end decoding throughput**. Our analysis shows that this is due to the large CPU overheads during decoding, which most likely come from Triton/CUDA kernel launch costs. These overheads **overlap** with the attention kernel execution, so even if the runtime of the attention kernel is greatly reduced, the CPU overheads still persist (as seen from the low GPU utilization), and thus there would be no throughput improvement.
> >
> > We also point out that even though our implementation saves memory IO, it does not perform explicit KV-cache eviction so it does not reduce **GPU memory consumption**. KV-cache eviction is technically possible but is very tricky to implement with our current implementation.
> >
> > Below we report the memory IO/FLOP savings and attention kernel runtime reduction during decoding for different model sizes and context lengths $L$. The prefill-length is $(7/8)L$ and we decode the last $L/8$ tokens. For 125M models we use 256k tokens per batch and for 360M and 760M models we use 128k tokens per batch.
> >
> > Memory IO/FLOP savings are shown in the following table. The savings are generally larger than those for pretraining likely because decoding happens at the end of the sequence so there are more KV entries that could pruned at the decoding stage.
> >
> > |             | 125M | 360M | 760M |
> > | :---------- | :--- | :--- | :--- |
> > | $L = 4096$  | 73%  | 73%  | 76%  |
> > | $L = 8192$  | 77%  | 76%  | 79%  |
> > | $L = 16384$ | 79%  | 77%  | 79%  |
> >
> > Attention runtime reduction are shown in the following table. Note we fix the number of tokens per batch and the runtime is for a single decoding step, so the runtime stays roughly constant across different context lengths.
> >
> > |             | 125M (256k tokens/batch) | 360M (128k tokens/batch) | 760M (128k tokens/batch) |
> > | :---------- | :----------------------- | :----------------------- | :----------------------- |
> > | $L = 4096$  | 60% (0.48 ms vs 1.19 ms) | 55% (0.38 ms vs 0.84 ms) | 62% (0.45 ms vs 1.18 ms) |
> > | $L = 8192$  | 63% (0.43 ms vs 1.18 ms) | 56% (0.37 ms vs 0.84 ms) | 64% (0.42 ms vs 1.18 ms) |
> > | $L = 16384$ | 63% (0.44 ms vs 1.18 ms) | 51% (0.41 ms vs 0.84 ms) | 62% (0.44 ms vs 1.18 ms) |
> >
> >
> >
> > We emphasize that these are preliminary results. For future work, reducing overheads such as Triton kernel launch costs or simply using a larger model (so that we won't be bottlenecked by CPU overheads) should allow us to translate the speedup of the attention kernel to throughput improvement.
> >
> >
> > > scalability is only tested up to 760 M parameters on a single 4 × L40S node, so behavior on multi-billion-parameter models and distributed settings remains unknown.
> >
> > This is indeed a limitation due to our limited computational resources. However, given that the improvements are fairly consistent across the three small scales we tested, it is reasonable to expect similar results for larger scales.
> >
> > > What fraction of total attention runtime is spent on the boundary-search algorithm at sequence length 16 k on L40S versus A100 GPUs
> >
> > Below we report the fraction of time spent on boundary search within the attention kernel for different context lengths and model sizes (i.e., boundary index search time divided by the total runtime of the attention kernel)
> >
> >
> > |             | 125M | 360M | 760m |
> > | :---------- | :--- | :--- | :--- |
> > | $L = 4096$  | 4%   | 3%   | 2%   |
> > | $L = 8192$  | 5%   | 4%   | 3%   |
> > | $L = 16384$ | 6%   | 4%   | 3%   |
> >
> >
> >
> > As shown in the table, the time spent on boundary index search is always below 6%. The result for A100 is similar (always below 7%). As mentioned in the General Response, the complexity of boundary index search is $O(\max(M, N))$ as opposed to $O(MN)$, where $M = L/B_q$ and $N=L/B_k$. In other words, the complexity is **linear w.r.t. the sequence length $L$** (note in Algorithm 1 we never reset $l$ to 1 in the for loop). This is negligible compared to the quadratic $O(L^2 d)$ costs of the actual attention computation.
> >
> > We hope our response addresses your concerns. If so, we would greatly appreciate it if you could raise your score. Feel free to let us know if you have more questions!

---

> > > ### Comment · Reviewer_KGU7 · 2025-06-05
> > >
> > > Thanks to the authors on the response on the overhead of boundary search and implementation details.

---

> > > > ### Author Response · Authors · 2025-06-05
> > > >
> > > > Thank you for confirming our clarifications! If your concerns are sufficiently addressed, we would greatly appreciate your consideration of a higher score. Please let us know if anything else needs attention.

---

### Author Response · Authors · 2025-06-03
**General Response**

We thank all reviewers for their time and valuable feedback on our paper. In this general response, we report some additional and updated results. We also elaborate on the computational costs of the boundary index search algorithm.

## Attention kernel runtime reduction

In the paper we reported (1) FLOP savings for the attention kernel and (2) training throughput improvements. Below we additionally report the speedup of attention kernel **in isolation, in terms of wall-clock time**, measured using `torch.cuda.Event`. This is more informative and reproducible than training throughput improvements, because training throughput also depends on non-attention components such as MLPs and RMSNorm. For example, with short context lengths (e.g., 4k) the attention kernel only accounts for a small proportion of the runtime of the entire model. So with short context lengths, even if ACP significantly reduces the runtime of the attention kernel, the improvement in training throughput may still be very small.

Below, we report the percentage reduction in runtime for the attention kernel across different model sizes and context lengths $L$. The actual runtimes, with and without ACP, are provided in parentheses. Each runtime measurement includes one forward and one backward pass, with a batch size of 32k tokens.

|             | 125M                      | 360M                      | 760M                       |
| :---------- | :------------------------ | :------------------------ | :------------------------- |
| $L = 4096$  | 51% (3.01 ms vs 6.13 ms)  | 54% (3.84 ms vs 8.27 ms)  | 58% (5.66 ms vs 13.31 ms)  |
| $L = 8192$  | 62% (4.39 ms vs 11.47 ms) | 62% (5.81 ms vs 15.48 ms) | 66% (8.45 ms vs 24.71 ms)  |
| $L = 16384$ | 68% (7.00 ms vs 21.63 ms) | 67% (9.74 ms vs 29.49 ms) | 70% (13.89 ms vs 46.12 ms) |


As shown in the Table, ACP consistently reduces the runtime of the attention kernel by 50% to 70%, corresponding to a two- to three-fold speedup.

## Updated training throughput results

We have optimized the attention kernel, along with other non-attention components such as RMSNorm. These changes significantly improve training throughput for models, both with and without ACP. Additionally, we fixed a benchmarking issue related to GPU power limits. The **relative** throughput improvements provided by ACP remain consistent with our previously reported results. The updated throughput improvement results are presented below. Actual throughput (in tokens per second) with and without ACP is shown in parentheses.

|             | 125M               | 360M              | 760M             |
| :---------- | :----------------- | :---------------- | :--------------- |
| $L = 4096$  | 12% (298k vs 267k) | 12% (110k vs 99k) | 12% (64k vs 57k) |
| $L = 8192$  | 24% (284k vs 228k) | 24% (105k vs 85k) | 23% (62k vs 50k) |
| $L = 16384$ | 45% (260k vs 179k) | 44% (96k vs 67k)  | 43% (57k vs 40k) |


Since we optimized the block sizes for the attention kernel, the pruning granularity has slightly changed, and thus the FLOP savings also have slightly changed. However, the difference from our reported results is extremely small (below 2% across all settings) so we omit the new FLOP saving results here.

## Clarification of the time complexity of the boundary index search algorithm

We emphasize the complexity of our boundary index search algorithm is $O(\max(M, N))$, where $M = L/B_q$ and $N=L/B_k$. In other words, the complexity is **linear with respect to the sequence length $L$** (note in Algorithm 1 we never reset $l$ to 1 inside the for loop). This is negligible compared to the quadratic $O(L^2 d)$ cost of the actual attention computation.

Below we report the percentage of wall clock time spent on boundary index search within the attention kernel (i.e., boundary search time divided by the total runtime of the attention kernel):

|             | 125M | 360M | 760M |
| :---------- | :--- | :--- | :--- |
| $L = 4096$  | 4%   | 3%   | 2%   |
| $L = 8192$  | 5%   | 4%   | 3%   |
| $L = 16384$ | 6%   | 4%   | 3%   |

As shown in the table, the boundary index search accounts for at most 6% of the total attention runtime.

---

### Decision · Program_Chairs · 2025-07-08

**Decision:**

Accept

**Comment:**

This paper proposes a method for improving the training efficiency of the Forgetting Transformer (FoX). The method leverages the special properties of the FoX decay bias matrix and dynamically prunes computations involving input-output dependencies that are strongly decayed by the forget gate. Experiments on language model pre-training with FoX demonstrate a significant reduction in FLOPs (up to 760M parameters and 16B tokens). The approach is technically sound, with provable guarantees, and the empirical results are solid—I do not consider the lack of larger-scale experiments a significant concern in this case. The authors have done a thorough job during the rebuttal, especially in addressing boundary index search and attention runtime, and all reviewers remained positive afterward. Therefore, I recommend acceptance of the paper.

Finally, I’m surprised that no reviewer has commented on the title of the paper. I believe it would be clearer to include terms like pre-training or training in the title, as the paper focuses on improving training efficiency. The word "pruning" may suggest an inference-time technique otherwise.